# WINDOW-BASED DISTRIBUTION SHIFT DETECTION FOR DEEP NEURAL NETWORKS

## ABSTRACT

To deploy and operate deep neural models in production, the quality of their predictions, which might be contaminated benignly or manipulated maliciously by input distributional deviations, must be monitored and assessed. Specifically, we study the case of monitoring the healthy operation of a deep neural network (DNN) receiving a stream of data, with the aim of detecting input distributional deviations over which the quality of the network's predictions is potentially damaged. Using selective prediction principles, we propose a distribution deviation detection method for DNNs. The proposed method is derived from a tight coverage generalization bound computed over a sample of instances drawn from the true underlying distribution. Based on this bound, our detector continuously monitors the operation of the network over a test window and fires off an alarm whenever a deviation is detected. This novel detection method consistently and significantly outperforms the state of the art with respect to the CIFAR-10 and ImageNet datasets, thus establishing a new performance bar for this task , while being substantially more efficient in time and space complexities.

## 1 INTRODUCTION

A wide range of artificial intelligence applications and services rely on deep neural models because of their remarkable accuracy. When a trained model is deployed in production, its operation should be monitored for abnormal behavior, and a flag should be raised if such is detected. Corrective measures can be taken if the underlying cause of the abnormal behavior is identified. For example, simple distributional changes may only require retraining with fresh data, while more severe cases may require redesigning the model (e.g., when new classes emerge).

In this paper we focus on distribution shift detection in the context of deep neural models and consider the following setting. Pretrained model $f$ is given, and we presume it was trained with data sampled from some distribution $P$. In addition to the dataset used in training $f$, we are also given an additional sample of data from $P$, which is used to train a detector $D$ (we refer to this as the detection-training dataset). While $f$ is used in production to process a stream of emerging input data, we continually feed $D$ with the most recent window $W_k$ of $k$ input elements. The detector also has access to the final layers of the model $f$ and should be able to determine whether the data contained in $W_k$ came from a distribution different from $P$. Detection algorithms based on a window, such as we consider here, have rarely been considered in the context of deep neural networks. To the best of our knowledge window-based deep detection has only been considered by (Rabanser et al., 2019). We emphasize that in this paper we are not considering the problem of identifying *single-instance* out-of-distribution or outlier instances (Liang et al., 2018; Hendrycks & Gimpel, 2017; Hendrycks et al., 2019; Golan & El-Yaniv, 2018; Ren et al., 2019; Nalisnick et al., 2019; Nado et al., 2021; Fort et al., 2021), but rather the information residing in a population of $k$ instances. Single-instance methods are trivially applicable to a window. However, these methods are not designed to detect population-based changes (see discussion in Section 2). We also note that this paper does not address the issue of characterizing the type of distribution shift, nor correcting it (by "redesigning" the model to make accurate predictions on the shifted distribution).

The detection of distribution shifts is a fundamental topic in machine learning and statistics, and the standard method for tackling it is by performing a dimensionality reduction over both the detection-training (source) and test (target) samples, and then applying a two-sample statistical test over these

reduced representations to detect a deviation. This is further discussed in Section 2. Distribution shift detection has been scarcely considered in the context of deep neural networks (DNNs). Deep models can benefit from the semantic representation created by the model itself, which provides meaningful dimensionality reduction that is readily available at the last layers of the model. Using the embedding layer (or softmax) along with statistical two-sample tests was recently proposed by (Lipton et al., 2018) and (Rabanser et al., 2019) who termed solutions of this structure black-box shift detection (BBSD). Using both the univariate Kolmogorov-Smirnov (KS) test and the maximum mean discrepancy (MMD) method, see details below, (Rabanser et al., 2019) achieve impressive detection results when using MNIST and CIFAR-10 as proxies for the distribution $P$. As we demonstrate here, the KS-BBSD method is also very effective over ImageNet when a stronger model is used (EfficientNet vs ResNet-18). BBSD methods have the disadvantage of being computationally intensive due to the use of two-sample tests between the detection-training set (which can, and are preferred to be the largest possible) and the window $W$ (a complexity analysis is provided in table 1).

We propose a different approach based on selective prediction (El-Yaniv & Wiener, 2010; Geifman & El-Yaniv, 2017), where a model quantifies its prediction uncertainty and abstains from predicting uncertain instances. First, we develop a method for selective prediction with guaranteed coverage. This method identifies the best abstaining threshold and coverage bound for a given pretrained classifier $f$, such that the resulting empirical coverage will not violate the bound with a high probability (when abstention is determined using the threshold). The guaranteed coverage method is of independent interest, and it is analogous to selective prediction with guaranteed risk (Geifman & El-Yaniv, 2017). Because the empirical coverage of such a classifier is highly unlikely to violate the bound if the underlying distribution remains the same, a systematic violation indicates a shift in distribution. To be more specific, given a detection-training sample $S_m$, our coverage-based detection algorithm computes $\log_2 m$ tight generalization coverage bounds, which are then used to detect a distribution shift in a window $W$ of test data. Due to its aggressive reduction of $S_m$ to $O(\log m)$ numbers, the proposed detection algorithm is extremely efficient in its computation requirements, unlike the baseline algorithms mentioned above, which follow the framework depicted in Figure 3 in Appendix 7.1. For example, consider the JFT-3B dataset (Zhai et al., 2021). Previous methods that require the processing of this set for each incoming window are infeasible, while our method allows one to summarize it with only 32 scalars.

In a comprehensive empirical study, we compared our coverage-based detection algorithm with the best-performing BBSD baselines, including the KS approach of (Rabanser et al., 2019). All methods used the same underlying models (ResNet-18, ResNet-50 and EfficientNet) for a fair comparison. We simulated source distributions using both the CIFAR-10 and ImageNet databases. Distribution shifts were produced using various methods, beginning with simple noise and ending with adversarial examples. Based on these experiments, we can claim that our coverage-based detection method is significantly more powerful than the baselines across a wide range of test window sizes.

To summarize, the contributions of this paper are: (1) A theoretically justified algorithm (Algorithm 1), that produces a coverage bound, which is of independent interest, and allows for the creation of selective classifiers with guaranteed coverage. (2) A theoretically motivated "windowed" detection algorithm (Algorithm 2), which detects a distribution shift over a window. (3) A comprehensive empirical study demonstrating significant improvements relative to existing baselines over a variety of datasets and architectures.

## 2 RELATED WORK

Distribution shift detection methods often comprise the following two steps: dimensionality reduction, and a two-sample test between the detection-training sample and test samples. In most cases, these methods are "lazy" in the sense that for each test sample, they make a detection decision based on a computation over the entire detection-training sample. Their performance will be sub-optimal if only a subset of the train sample is used. Figure 3 in Appendix 7.1 illustrates this general framework.

The use of dimensionality reduction is optional. It can often improve performance by focusing on a less noisy representation of the data. Dimensionality reduction techniques include no reduction, *principal components analysis* (Wold et al., 1987), *sparse random projection* (Bingham & Mannila, 2001), *autoencoders* (Rumelhart et al., 1985; Pu et al., 2016), *domain classifiers*, (Rabanser et al., 2019) and more. In this work we focus on *black box shift detection* (BBSD) methods (Lipton et al., 2018), that rely on deep neural representations of the data generated by a pretrained model. The representation we extract from the model will typically utilize either the softmax outputs

(acronymed **BBSD-S**) or the embeddings (acronymed **BBSD-E**). Due to the dimensionality of the final representation, multivariate or multiple univariate two-sample tests can be conducted.

By combining BBSD-S with a Kolmogorov-Smirnov (**KS**) statistical test (Massey Jr, 1951) and using the Bonferroni correction (Bland & Altman, 1995), (Rabanser et al., 2019) achieved state-of-the-art results in distribution shift detection in the context of image classification (MNIST and CIFAR-10). We acronym their method as **KS-BBSD-S**. The *univariate* KS test processes individual dimensions separately; its statistic is calculated by computing the largest difference $Z$ of the *cumulative density functions* (CDFs) across all dimensions as follows: $Z = \sup_{z} |F_P(z) - F_Q(z)|$, where $F_Q$ and $F_P$ are the empirical CDFs of the detection-training and test data (which are sampled from $P$ and $Q$, respectively; see Section 3). The Bonferroni correction rejects the null hypothesis when the minimal p-value among all tests is less than $\frac{\alpha}{d}$, where $\alpha$ is the significance level of the test, and $d$ is the number of dimensions. Although there have been several less conservative approaches to aggregation (Heard & Rubin-Delanchy, 2018; Loughin, 2004), they usually assume some dependencies among the tests.

The *maximum mean discrepancy* (**MMD**) method (Gretton et al., 2012) is a kernel-based multivariate test that can be used to distinguish between probability distributions $P$ and $Q$. Formally, $MMD^2(\mathcal{F}, P, Q) = ||\boldsymbol{\mu_P} - \boldsymbol{\mu_Q}||^2_{\mathcal{F}^2}$, where $\boldsymbol{\mu_P}$ and $\boldsymbol{\mu_Q}$ are the mean embeddings of $P$ and $Q$ in a reproducing kernel Hilbert space $\mathcal{F}$. Given a kernel $\mathcal{K}$, and samples, $\{x_1, x_2, \ldots, x_m\} \sim P^m$ and $\{x'_1, x'_2, \ldots, x'_k\} \sim Q^k$, an unbiased estimator for $MMD^2$ can be found in (Gretton et al., 2012; Serfling, 2009). (Sutherland et al., 2017) and (Gretton et al., 2012) used the RBF kernel $\mathcal{K}(x, x') = e^{-\frac{1}{2\sigma^2}||x-x'||^2_2}$, where $2\sigma^2$ is set to the median of the pairwise Euclidean distances between all samples. By performing a permutation test on the kernel matrix, the p-value is obtained. In our experiments (see Section 5), we thus use four baselines: KS-BBSD-S, KS-BBSD-E, MMD-BBSD-S, and MMD-BBSD-E.

As mentioned in the introduction, our work is complementary to the topic of single-instance out-of-distribution (OOD) detection (Liang et al., 2018; Hendrycks & Gimpel, 2017; Hendrycks et al., 2019; Golan & El-Yaniv, 2018; Ren et al., 2019; Nalisnick et al., 2019; Nado et al., 2021; Fort et al., 2021). Obviously, these methods can be applied in a trivial manner over a window, by applying the detector to each instance in the given window. However, these methods typically do not consider the population statistics over the window. Interestingly, we demonstrate in Section 5.1, Figure 2 that near perfect performance can be achieved by (our) window-based detection when the window is sufficiently large. For instance, a detection AUROC score close to $100\%$ can be obtained over a 1K window when considering various distributional changes such as adversarial attacks, Gaussian noise, and more. OOD detection methods rarely achieve such a near-perfect AUROC score when considering such distributional changes.

Finally, we mention (Geifman & El-Yaniv, 2017) who developed a risk generalization bound for selective classifiers (El-Yaniv & Wiener, 2010). The bound presented in that paper is analogous to the coverage generalization bound we present in Theorem 4.2. The risk bound in (Geifman & El-Yaniv, 2017) can also be used for shift-detection. To apply their risk bound to this task, however, labels, which are not available, are required. Our method (Section 4) detects distribution shifts without using any labels.

## 3 PROBLEM FORMULATION

We consider the problem of detecting distribution shifts in input streams provided to pretrained deep neural models. Let $\mathcal{P} \triangleq P_X$ denote a probability distribution over an input space $\mathcal{X}$, and assume that a model $f$ has been trained on a set of instances drawn from $\mathcal{P}$. Samples drawn from $\mathcal{P}$ are referred to as *in-distribution* (ID), or *detection-training* data. Consider a setting where the model $f$ is deployed and while being used in production its input distribution might change or even be attacked by an adversary. Our goal is to detect such events to allow for appropriate action, e.g., retraining the model with respect to the revised distribution.

Inspired by (Rabanser et al., 2019), we formulate this problem as follows. We are given a pretrained model $f$ (whose ID training data was sampled from $\mathcal{P}$). Then having $f$ and additional ID detection-training data $S_m \sim \mathcal{P}^m$ (possibly unlabeled), we would like to train a detection model to be able to detect a distribution shift; namely, discriminate between windows containing ID data, and *alternative-*

*distribution* (AD) data. Thus, given an unlabeled test sample window $W_k \sim Q^k$, where $Q$ is a possibly different distribution, the objective is to determine whether $\mathcal{P} \neq Q$. We also ask what is the smallest test sample size $k$ required to determine that $\mathcal{P} \neq Q$. Since typically the training set $S_m$ can be quite large, we further ask whether it is possible to devise an effective detection procedure whose time complexity is $o(m)$.

## 4 PROPOSED METHOD – COVERAGE-BASED DETECTION

In this section we present a novel technique for detecting a distribution shift based on selective prediction principles (definitions follow). We develop a tight generalization coverage bound, based on ID training set sampled i.i.d. from the source distribution. This bound should hold with high probability for ID data from the source distribution. For a given window of data at test time we calculate its empirical coverage, and compare it to the theoretical coverage bound. Since the bound should hold with high probability on ID data, a coverage violation indicates w.h.p. a distribution shift from the ID source.

### 4.1 SELECTION WITH GUARANTEED COVERAGE

We begin by introducing basic selective prediction terminology and definitions that are required to describe our method. Consider a standard multiclass classification problem, where $\mathcal{X}$ is some feature space (e.g., raw image data) and $\mathcal{Y}$ is a finite label set, $\mathcal{Y} = \{1, 2, 3, ..., C\}$, representing $C$ classes. Let $P(X, Y)$ be a probability distribution over $\mathcal{X} \times \mathcal{Y}$, and define a *classifier* as a function $f : \mathcal{X} \to \mathcal{Y}$. We refer to $P$ as the *source distribution*. A *selective classifier* (El-Yaniv & Wiener, 2010) is a pair $(f, g)$, where $f$ is a classifier and $g : \mathcal{X} \to \{0, 1\}$ is a *selection function* (El-Yaniv et al., 2010), which serves as a binary qualifier for $f$ as follows,

$$(f, g)(x) \triangleq \begin{cases} f(x), & \text{if } g(x) = 1; \\ \text{don't know}, & \text{if } g(x) = 0. \end{cases}$$

A general approach for constructing a selection function based on a given classifier $f$ is to work in terms of a *confidence-rate function* (Geifman et al., 2019), $\kappa_f : \mathcal{X} \to \mathbb{R}^+$, referred to as CF. The CF $\kappa_f$ should quantify confidence in predicting the label of $x$ based on signals extracted from $f$ (Geifman et al., 2019). The most common and well-known CF for a classification model $f$ (with softmax at its last layer) is its *softmax response* (SR) value (Cordella et al., 1995; De Stefano et al., 2000; Hendrycks & Gimpel, 2017). A related CF is the entropy of the softmax vector (more precisely one minus the entropy, as $\kappa_f$ should indicate confidence), denoted here as *Soft-Entropy* (Gal, 2016). A given CF $\kappa_f$ can be straightforwardly used to define a selection function: $g_\theta(x) \triangleq g_\theta(x|\kappa_f) = \mathbb{1}[\kappa_f(x) \geq \theta]$, where $\theta$ is a user-defined constant. For any selection function, we define its *coverage* w.r.t. a distribution $P$, and its *empirical coverage* w.r.t. a sample $S_k \triangleq \{x_1, x_2, \ldots x_k\}$, as $c(\theta, P) \triangleq \mathbb{E}_P[g_\theta(x)]$, and $\hat{c}(\theta, S_k) \triangleq \frac{1}{k} \sum_{i=1}^{k} g_\theta(x_i)$, respectively.

Given a bound on the expected coverage for a given selection function, we can use it to detect a distribution shift via violations of the bound. We now develop such a bound and show how to use it to detect distribution shifts. For a classifier $f$, a detection-training sample $S_m \sim P^m$, a confidence parameter $\delta > 0$, and a desired coverage $c^* > 0$, our goal is to use $S_m$ to find a $\theta$ value (which implies a selection function $g_\theta$) that guarantees the desired coverage. This means that *under coverage* should occur with probability of at most $\delta$,

$$\mathbf{Pr}_{S_m}\{c(\theta, P) < c^*\} < \delta. \tag{1}$$

A $\theta$ that guarantees Equation (1) provides a probabilistic lower bound, guaranteeing that coverage $c$ of ID unseen population (sampled from $P$) satisfies $c > c^*$ with probability of at least $1 - \delta$. A symmetric upper bound is presented in Appendix 7.5.

We now describe the *selection with guaranteed coverage* (SGC) algorithm. The algorithm receives as input a classifier $f$, a CF $\kappa_f$, a confidence parameter $\delta$, a target coverage $c^*$, and a detection-training set $S_m$. The algorithm performs a binary search to find the optimal coverage lower bound with confidence $\delta$, and outputs a coverage bound $b^*$ and the threshold $\theta$, defining the selection function. A pseudo code of the SGC algorithm appears in Algorithm 1. Our analysis of the SGC algorithm makes

---

**Algorithm 1:** *Selection with guaranteed coverage* **(SGC)**

---

1  **Input:** train set: $S_m$, confidence-rate function: $\kappa_f$, confidence parameter $\delta$, target coverage: $c^*$.
2  Sort $S_m$ according to $\kappa_f(x_i)$, $x_i \in S_m$ (and now assume w.l.o.g. that indices reflect this ordering).
3  $z_{\min} = 1$, $z_{\max} = m$
4  **for** $i = 1$ **to** $k = \lceil \log_2 m \rceil$ **do**
5  |    $z = \lceil (z_{\min} + z_{\max})/2 \rceil$
6  |    $\theta_i = \kappa_f(x_z)$
7  |    Calculate $\hat{c}_i(\theta_i, S_m)$
8  |    Solve for $b_i^*(m, m \cdot \hat{c}_i(\theta_i, S_m), \frac{\delta}{k})$ {see Lemma 4.1}
9  |    **if** $b_i^*(m, m \cdot \hat{c}_i(\theta_i, S_m), \frac{\delta}{k}) \leq c^*$ **then**
10 |    |    $z_{\max} = z$
11 |    **else**
12 |    |    $z_{\min} = z$
13 |    **end if**
14 **end for**
15 **Output:** bound: $b_k^*(m, m \cdot \hat{c}_k(\theta_k, S_m), \frac{\delta}{k})$, threshold: $\theta_k$.

---

use of Lemma 4.1, which gives a tight numerical (generalization) bound on the expected coverage, based on a test over a sample. The proof of Lemma 4.1 is nearly identical to Langford's proof of Theorem 3.3 in (Langford & Schapire, 2005), p. 278, where instead of the empirical error used in (Langford & Schapire, 2005), we use the empirical coverage, which is also a Bernoulli random variable. To gain a better understanding of Lemma 4.1, please see Appendix 7.8.

**Lemma 4.1.** *Let P be any distribution and consider a selection function $g_\theta$ with a threshold $\theta$ whose coverage is $c(\theta, P)$. Let $0 < \delta < 1$ be given and let $\hat{c}(\theta, S_m)$ be the empirical coverage w.r.t. the set $S_m$, sampled i.i.d. from P. Let $b^*(m, m \cdot \hat{c}(\theta, S_m), \delta)$ be the solution of the following equation:*

$$\arg\min_b \left( \sum_{j=0}^{m \cdot \hat{c}(\theta, S_m)} \binom{m}{j} b^j (1-b)^{m-j} \leq 1 - \delta \right). \tag{2}$$

*Then,*
$$\mathbf{Pr}_{S_m}\{c(\theta, P) < b^*(m, m \cdot \hat{c}(\theta, S_m), \delta)\} < \delta. \tag{3}$$

The following is a uniform convergence theorem for the SGC procedure stating that all the calculated bounds are valid simultaneously with a probability of at least $1 - \delta$. More specifically, we apply Lemma 4.1 $\lceil \log_2 m \rceil$ times (using a binary search), and the returned value (the last one) depends on all the other values. Since these applications are dependent we must use the union bound.

**Theorem 4.2.** *(SGC – Uniform convergence) Assume $S_m$ is sampled i.i.d. from P, and consider an application of Algorithm 1. For $k = \lceil \log_2 m \rceil$, let $b_i^*(m, m \cdot \hat{c}_i(\theta_i, S_m), \frac{\delta}{k})$ and $\theta_i$ be the values obtained in the $i^{th}$ iteration of Algorithm 1. Then,*

$$\mathbf{Pr}_{S_m}\{\exists i : c(\theta_i, P) < b_i^*(m, m \cdot \hat{c}_i(\theta_i, S_m), \frac{\delta}{k})\} < \delta.$$

*Proof (sketch - see full proof in the Appendix 7.2.1).* Define,
$\mathcal{B}_{\theta_i} \triangleq b_i^*(m, m \cdot \hat{c}_i(\theta_i, S_m), \frac{\delta}{k})$, $\mathcal{C}_{\theta_i} \triangleq c(\theta_i, P)$, then,

$$
\begin{aligned}
\mathbf{Pr}_{S_m}\{\exists i : \mathcal{C}_{\theta_i} < \mathcal{B}_{\theta_i}\} &= \sum_{i=1}^{k} \int_0^1 d\theta' \mathbf{Pr}_{S_m}\{\mathcal{C}_{\theta'} < \mathcal{B}_{\theta'}\} \cdot \mathbf{Pr}_{S_m}\{\theta_i = \theta'\} \\
&< \sum_{i=1}^{k} \int_0^1 d\theta' \frac{\delta}{k} \cdot \mathbf{Pr}_{S_m}\{\theta_i = \theta'\} = \sum_{i=1}^{k} \frac{\delta}{k} = \delta.
\end{aligned}
$$

$\square$

## 4.2   Coverage-Based Detection Algorithm

Our detection algorithm works by applying SGC $\lfloor \log_2 m \rfloor$ times for various target coverage values ($c^*$). Application $j$ of SGC, with $c_j^*$, yields a corresponding pair, $(b_j^*, \theta_j)$, of a bound and a threshold,

respectively. All applications of SGC are over the same sample $S_m \sim P^m$. In our experiments the $\lfloor \log_2 m \rfloor$ target coverages are uniformly spread in the interval $[0, 1]$ (excluding its end points 0 and 1). Recalling that $k$ is the size of the window sample, $W_k \sim Q^k$, define,

$$
\mu \triangleq \frac{1}{\lfloor \log_2 m \rfloor} \sum_{j=1}^{\lfloor \log_2 m \rfloor} b_j^*,
$$

$$
\hat{\mu} \triangleq \frac{1}{\lfloor \log_2 m \rfloor} \sum_{j=1}^{\lfloor \log_2 m \rfloor} \hat{c}_j(\theta_j, W_k) = \frac{1}{\lfloor \log_2 m \rfloor} \sum_{j=1}^{\lfloor \log_2 m \rfloor} \frac{1}{k} \sum_{i=1}^{k} g_{\theta_j}(x_i)
$$

$$
= \frac{1}{k \lfloor \log_2 m \rfloor} \sum_{j=1}^{\lfloor \log_2 m \rfloor} \sum_{i=1}^{k} g_{\theta_j}(x_i).
$$

If $Q$ is identical to $P$ (i.e., no distribution shift), we expect that the bound computed by SGC over $S_m$ will hold over $W_k$ as well; namely, $\hat{c}_j(\theta_j, W_k) \geq b_j^*$, for every iteration $j$ of SGC.

As $\hat{\mu}$ represents the average of $k \lfloor \log_2 m \rfloor$ values, and as in our experiments, $k \lfloor \log_2 m \rfloor \gg 30$, a well known rule of thumb (James et al. (2013), p. 67, states that $\hat{\mu}$ is nearly normally distributed. Therefore, we can apply a t-test[1] to accept or reject the null hypothesis $H_0 : \hat{\mu} \geq \mu$, where the alternative hypothesis is $H_1 : \hat{\mu} < \mu$. The null hypothesis is rejected if the p-value is less than the desired threshold (significance level) of the test $\alpha$ (user defined). Thus, when evaluating performance using any tradeoff-based metric (e.g., AUROC) we vary $\alpha$ to obtain the tradeoff. A pseudo code of our *coverage-based detection algorithm* appears in Algorithm 2.

---

**Algorithm 2: *Coverage-Based Detection***

---

1   // Training
2   **Input Training:** $\delta, \kappa_f, \{c_j^*\}_{j=1}^{\lfloor \log_2 m \rfloor}$
3   **for** $j = 1$ **to** $k = \lfloor \log_2 m \rfloor$ **do**
4     $\big|$   $b_j^*, \theta_j = \text{SGC}(S_m, \delta, c_j^*, \kappa_f)$
5   **end for**
6   $\mu \triangleq \frac{1}{\lfloor \log_2 m \rfloor} \sum_{j=1}^{\lfloor \log_2 m \rfloor} b_j^*$
7   **Output Training:** $\mu, \{(b_j^*, \theta_j)\}_{j=1}^{\lfloor \log_2 m \rfloor}$
8   // Detection model
9   **Input Detection:** $\mu, \{(b_j^*, \theta_j)\}_{j=1}^{\lfloor \log_2 m \rfloor}, \kappa_f, \alpha, k$ **while** True **do**
10     Receive windows $W_k = \{x_1, x_2, \ldots, x_k\}$
11     **for** $j = 1$ **to** $k = \lfloor \log_2 m \rfloor$ **do**
12       $\big|$   $\hat{c}_j(\theta_j, W_k) \triangleq \frac{1}{k} \sum_{i=1}^{k} g_{\theta_j}(x_i)$
13     **end for**
14     $\hat{\mu} \triangleq \frac{1}{k \lfloor \log_2 m \rfloor} \sum_{j=1}^{\lfloor \log_2 m \rfloor} \sum_{i=1}^{k} g_{\theta_j}(x_i)$
15     Obtain p-value from t-test, $H_0 : \hat{\mu} \geq \mu, H_1 : \hat{\mu} < \mu$
16     **if** $p_{value} < \alpha$ **then**
17       Shift_detected $\leftarrow$ True
18       **Output Detection:** Shift_detected, $p_{\text{value}}$
19     **end if**
20   **end while**

---

We train only once using SGC (Algorithm 1) on the detection-training data $S_m$ for $\lfloor \log_2 m \rfloor$ times, in order to construct the pairs $\{b_j^*, \theta_j\}_{j=1}^{\lfloor \log_2 m \rfloor}$. Our detection model utilizes these pairs to monitor a given model, receiving at each time instant a test sample window of size $k$ (user defined), $W_k = \{x_1, x_2, \ldots, x_k\}$, which is checked to see if its content is distributionally shifted from the underlying distribution reflected by the detection-training data $S_m$. A schematic diagram of our procedure on window data, $W_k$, appears in Figure 1.

In comparison to the previous baselines, our proposed method is extremely more efficient, as shown in table 1. A derivation of the complexity bounds can be found in Appendix 7.3.

---

[1]In our experiments we apply SciPy's stats.ttest_1samp t-test implementation (Virtanen et al., 2020).

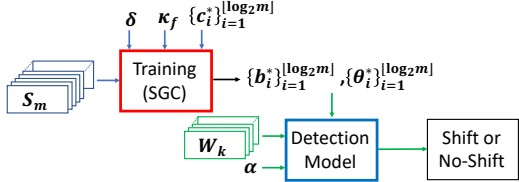

Figure 1: Our detection procedure comprising two stages, training and detecting. The detection stage requires parameters $\{b_i^*\}_{i=1}^{\lfloor \log_2 m \rfloor}$ and $\{\theta_i\}_{i=1}^{\lfloor \log_2 m \rfloor}$ (computed at the training stage), i.e., only $O(\log_2 m)$ parameters relative to the training data size ($m$).

| Detection Method | Space | Time |
|---|---|---|
| **Coverage-Based Detection** (Ours) | $O(k + \log m)$ | $O(k + \log m)$ |
| **MMD** | $O(m^2 + k^2 + mk)$ | $O\left(d(m^2 + k^2 + mk)\right)$ |
| **KS** | $O\left(d(m + k)\right)$ | $O\left(d(m \log m + k \log k)\right)$ |

Table 1: Complexity comparison. **Orange bolds** entries indicate the best detection complexity. $m$, $k$ refers to the detection-training size, and window size, respectively. $d$ refers to the number of dimensions after dimensionality reduction. A derivation of the complexity bounds can be found in Appendix 7.3.

## 5 EMPIRICAL STUDY

In this section we evaluate the performance of our coverage-based detection algorithm, as well as the baselines defined in Section 2. Three experiments are conducted. The first experiment, is carried out using synthetic data, and its purpose is to provide some insight into the operation of our algorithm. Specifically, we use synthetic data in order to demonstrate that our proposed bound tightly holds, and when a distribution shift occurs, it is violated as expected. Unfortunately, due to lack of space, this experiment is described in Appendix 7.4.1. The purpose of the other two experiments is to examine the performance of our detection algorithm, with respect to the four baselines (KS-BBSD-S, KS-BBSD-E, MMD-BBSD-S, and MMD-BBSD-E; see Section 2). These experiments are carried out on the CIFAR-10 (Krizhevsky et al., 2009) and the ImageNet (Deng et al., 2009) datasets, which are used as our detection-training (ID) data. As our test data we use a number of datasets and corruptions of the detection-training data to represent a variety of distribution shifts (see details below).

We now define the metrics we use to evaluate detection performance. Following (Liang et al., 2018), we consider a soft binary classification setting (is $P = Q$ or $P \neq Q$, see Section 3), where the answer is determined based on a decision threshold. We thus use the *Area Under an Operating Characteristic curve* (AUROC) and the *Area under the Precision-Recall curve* (AUPR).

**AUROC** is a threshold-independent metric (Davis & Goadrich, 2006). The ROC curve depicts the relationship between the *true positive rate* and the *false positive rate*. The AUROC can be interpreted as the probability that a positive labeled window will have a higher detection score than a negative one (Fawcett, 2006). An AUROC score of $100\%$ corresponds to a perfect detector. AUPR is also a threshold-independent metric. The precision-recall (PR) curve is a function giving the relation between the precision = TP/(TP+FP) and the recall = TP/(TP+FN) for all threshold values. Following (Liang et al., 2018), we separate this metric into **AUPR-Tr** and **AUPR-Te**, which measure the area under the PR curve at which positive test windows are containing ID (detection-training) data (AUPR-Tr) and AD (test) data (AUPR-Te), respectively. It should be noted that AUPR-Te is almost saturated in most of our experiments (due to data imbalance), however, we decided to include the results of AUPR-Te nonetheless for completeness.

### 5.1 CIFAR-10 AND IMAGENET

We now present our primary empirical study using the CIFAR-10 (Krizhevsky et al., 2009) and ImageNet (Deng et al., 2009) datasets. In both cases the customary validation/test sets (10,000 instances in CIFAR-10 and 50,000 in ImageNet) are split randomly (and uniformly) 30 times into two groups to form 30 detection-training sets ($S_m$, see Section 3) and tests pairs, that are used to train the detection model and evaluate its performance, respectively. The size of the test set is 1000 in all cases. Thus, the sizes of detection-training sets and test sets are (9000, 1000) for CIFAR-10 and (49,000, 1000) for ImageNet. Detection models are challenged by forming windows $W_k$, for various sizes of $0 < k \leq 1000$, where in each case $W_k$ comprises simulated distribution shifts that represent various

distributions $Q$, including the no-shift case (where $Q = P$), to check for false-alarms (see details below). To aggregate a metric score such as AUROC, over all window sizes to a single number, we define the metrics Agg-AUROC (see Figure 2-left), Agg-AUPR-Tr, and Agg-AUPR-Te, which are used in Tables 2, 3. These metrics denote the area under the curve of the metric as a function of $k$. We also display the scores individually for each window size, see details later.

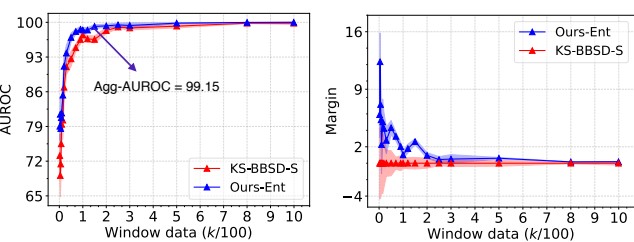

Figure 2: **ImageNet Experiments**. AUROC as a function of the window size $k$ (left), and the margin between our best model (Ours-Ent), and the best baseline, KS-BBSD-S (right). The margin is the difference between the AUROC scores of Ours-Ent and KS-BBSD-S. One-$\sigma$ error-bars are shadowed.

All figures show metric values with a one-$\sigma$ error-bar obtained via bootstrapping. Throughout all experiments, we use the coverage-based detection algorithm (Algorithm 2) with $\delta = 0.001$, and in the applications described below, we instantiate CF ($\kappa_f$) with both SR and Soft-Entropy. These two applications of our method are referred to as **Ours-SR** and **Ours-Ent**.

### 5.1.1 CIFAR-10 EXPERIMENTS

To evaluate performance, we used two pretrained ResNet-18 and ResNet-50 (He et al., 2016). Following (Rabanser et al., 2019) we simulated the following shifts: **Rotations**: with angles $\theta \in \{10, 40, 90\}$, **Gaussian noise**: with STD $\sigma \in \{\frac{40}{255}, \frac{60}{255}, \frac{80}{255}\}$, the **CIFAR-100** dataset (Krizhevsky et al., 2009), the **SVHN** dataset (Netzer et al., 2011), and **Adversarial**: *fast gradient sign method* (Goodfellow et al., 2015), *projected gradient descent* (Madry et al., 2018), *Carlini Wagner* (Carlini & Wagner, 2017), referred to as FGSM, PGD and CW, respectively. The aggregated results for all these shifts are summarized in Table 2.

| **Method** | | **ResNet-18** | | **ResNet-50** | |
|---|---|---|---|---|---|
| | | Agg-AUROC | Agg-AUPR-Te/Tr | Agg-AUROC | Agg-AUPR-Te/Tr |
| **Ours** | **Ent** | **98.57** | **99.85** / **92.18** | **98.72** | **99.87** / **92.08** |
| | SR | 98.56 | **99.85** / 92.11 | 98.7 | **99.87** / 91.84 |
| **KS** | BBSD-S | 97.27 | 99.73 / 82.58 | 97.61 | 99.77 / 83.71 |
| | BBSD-E | 96.76 | 99.69 / 77.09 | 97.47 | 99.75 / 82.94 |
| **MMD** | BBSD-S | 84.3 | 98.38 / 27.75 | 83.13 | 98.19 / 26.98 |
| | BBSD-E | 87.25 | 98.73 / 35.1 | 85.7 | 98.47 / 32.09 |

Table 2: **CIFAR-10 Experiments**. An underlined entry indicates the best baseline among the proposed baselines (KS-BBSD-S, KS-BBSD-E, MMD-BBSD-S, and MMD-BBSD-E, see Section 2), and **bold orange** indicates the best detection score for a given model and metric.

Orange bold entries indicate the best evaluation score, while underlined entries indicate the best evaluation score vis-a-vis the baselines. It is evident that Ours-Ent dominates all methods w.r.t. all performance metrics, and when applied with both architectures. Interestingly, both our detection variants excel in the Agg-AUPR-Tr metric, compared to the baselines. This reflects the fact that our detection methods have significantly fewer false-alarms compared to the baselines; i.e., the baselines are more likely to misidentify ID windows as AD windows than we are. This is crucial when considering distribution shift detection, since most of the time the model is processing ID data. Among the baselines, KS-BBSD-S dominates the three other baselines, in line with the findings of (Rabanser et al., 2019), who also experimented over CIFAR-10. For individual results regarding each window size, see Appendix 7.4.2.

### 5.1.2 IMAGENET EXPERIMENTS

For the ImageNet dataset, we simulated a larger number of shifts than we did in the CIFAR-10 experiments (Section 5.1.1): **Rotations**: $\theta \in \{10, 40, 90, 120, 150, 180\}$, the **ImageNet-O** dataset

(Hendrycks et al., 2021), the **ImageNet-A** dataset (Hendrycks et al., 2021), and the **ImageNet-C** dataset (Hendrycks & Dietterich, 2019), which contains 75 common visual distortions with intensities ranging from 1 to 5; for our experiments we use $\{1, 3, 5\}$. We also consider **adversarial attacks**: *fast gradient sign method* (Goodfellow et al., 2015), *projected gradient descent* (Madry et al., 2018), *Carlini Wagner* (Carlini & Wagner, 2017), referred to as FGSM, PGD and CW, respectively. For each distribution shift type we considered three proportions $p$ of window contamination, where $p$ denotes the proportion of shifted instances in the window. For example, if $p = \frac{1}{3}$, a third of the window is contaminated with instances sampled from the shifted distribution, while the remaining instances are sampled from $P$. In our experiments we used proportions $p \in \{1, \frac{2}{3}, \frac{1}{3}\}$. Due to lack of space we present only the most successful baseline (we experimented with all), namely, the KS-BBSD-S detection model, which was also the most successful baseline regarding the CIFAR-10 experiment. For this ImageNet experiment, we used the strong EfficientNet-B0 network (Tan & Le, 2019) as the underlying model for our methods and the baseline. We took a model that was pretrained over ImageNet (Wightman, 2019). In Table 3 we summarize the aggregate results for all these shifts.

| Method | | EfficientNet-B0 | | | | | |
| | | $p = 1$ | | $p = 2/3$ | | $p = 1/3$ | |
| | | Agg-AUROC | Agg-AUPR-Te/Tr | Agg-AUROC | Agg-AUPR-Te/Tr | Agg-AUROC | Agg-AUPR-Te/Tr |
|---|---|---|---|---|---|---|---|
| **Ours** | **Ent** | **99.15** | **99.93** / 94.21 | **98.23** | **99.84** / 87.21 | **93.50** | **99.45** / **61.76** |
| | SR | 99.12 | 99.92 / **94.29** | 98.18 | **99.84** / 86.21 | 93.38 | 99.44 / 59.95 |
| KS | BBSD-S | 98.3 | 99.86 / 83.85 | 96.06 | 99.67 / 70.57 | 89.16 | 99.02 / 44.28 |

Table 3: **ImageNet Experiments**. **Orange bolds** indicate the best detection score for a given metric, and contamination percentage ($p$). KS-BBSD-S (Rabanser et al., 2019), is the Kolmogorov-Smirnov (KS) statistical test (Massey Jr, 1951), using a Black Box Shift Detection (BBSD) (Lipton et al., 2018) method, using Softmax (S); namely, KS-BBSD-S. According to (Rabanser et al., 2019), and based on our analysis in Table 2, this is the best proposed baseline.

Bold orange entries indicate the highest evaluation scores. In line with the results of the CIFAR-10 experiment (Section 5.1.1), Ours-Ent dominates both other methods (Ours-SR, KS-BBSD-S) for all performance metrics, and for each contamination percentage ($p$) considered, with the exception of Agg-AUPR-Tr ($p = 1$), where in this case Ours-SR dominates. A significant margin separates Ours-Ent from KS-BBSD-S, and this margin becomes even larger when the contamination ($p$) decreases (see Appendix 7.4.3). In Figure 2(left), one can see the resulting AUROC scores as a function of the window size $k$ for $p = 1$. In Figure 2(right) we also display the margin between Ours-Ent and the best baseline, KS-BBSD-S (the rest of the metrics, appear in the Appendix, Figure 8). For the vast majority of window sizes, Ours-Ent performs significantly better than the best baseline, KS-BBSD-S; Furthermore, see Appendix 7.4.3, Ours-Ent consistently outperforms KS-BBSD-S over all metrics, across all percentages of affected data ($p \in \{1, \frac{2}{3}, \frac{1}{3}\}$), and for all window sizes, where the biggest gap is over the AUPR-Tr metric, which accounts for lower false-alarms compared to the baselines. We conclude that in our setting, Soft-Entropy is the best CF. Appendix 7.6 provides a detailed comparison.

## 6 CONCLUDING REMARKS

We presented a novel and powerful method for the detection of distribution shifts within a given window of samples. This coverage-based detection algorithm is theoretically motivated and can be applied to any pretrained model. Due to its low computational complexity, our method, unlike typical baselines, is practicable. Our comprehensive empirical studies demonstrate that the proposed method works very well, and overall significantly outperforms the baselines on both the CIFAR-10 and ImageNet datasets, across a number of neural architectures and a variety of distribution shifts, including adversarial examples. In addition, our coverage bound is of independent interest and allows for the creation of selective classifiers with guaranteed coverage. Several directions for future research are left open. Although we only considered classification, our method can be extended to regression using an appropriate confidence-rate function such as the MC-dropout (Gal & Ghahramani, 2016). Extensions to other tasks, such as object detection and segmentation, would be very interesting. It would also be interesting to examine other types of shift benchmarks such as (Koh et al., 2021). In our method, the information from the multiple coverage bounds was aggregated by averaging, but it is plausible that other statistics or weighted averages could provide more effective detections. Finally, an interesting open question is whether one can benefit from using outlier/adversarial detection techniques combined with population-based detection techniques (as discussed here).

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

# 7 APPENDIX

## 7.1 SHIFT-DETECTION GENERAL FRAMEWORK

The general framework for shift-detection can be found in the following figure, Figure 3.

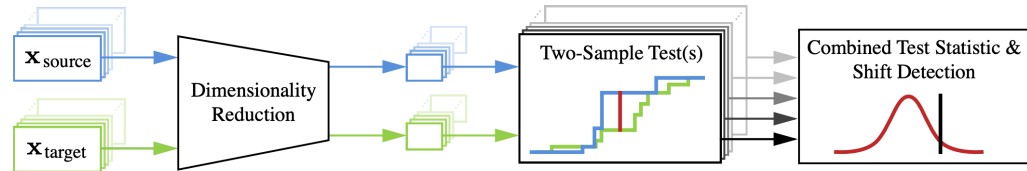

Figure 3: The procedure of detecting a dataset shift using dimensionality reduction and then a two-sample statistical test. The dimensionality reduction is applied to both the detection-training (source) and test (target) data, prior to being analyzed using statistical hypothesis testing. This figure is taken from (Rabanser et al., 2019).

## 7.2 PROOFS

### 7.2.1 PROOF FOR THEOREM 4.2

*Proof.* Define

$$\mathcal{B}_{\theta_i} \triangleq b_i^*(m, m \cdot \hat{c}_i(\theta_i, S_m), \frac{\delta}{k}),$$

$$\mathcal{C}_{\theta_i} \triangleq c(\theta_i, P).$$

Consider the i[th] iteration of SGR over a detection-training set $S_m$, and recall that, $\theta_i = \kappa_f(x_z)$, $x_z \in S_m$ (see Algorithm 1). Therefore, $\theta_i$ is a random variable (between zero and one), since it is a function of a random variable ($x \in S_m$). Let $\mathbf{Pr}_{S_m}\{\theta_i = \theta'\}$ be the probability that $\theta_i = \theta'$.

Therefore,

$$\mathbf{Pr}_{S_m}\{\mathcal{C}_{\theta_i} < \mathcal{B}_{\theta_i}\}$$
$$= \int_0^1 d\theta' \mathbf{Pr}_{S_m}\{\mathcal{C}_{\theta_i} < \mathcal{B}_{\theta_i}|\theta_i = \theta'\} \cdot \mathbf{Pr}_{S_m}\{\theta_i = \theta'\}$$
$$= \int_0^1 d\theta' \mathbf{Pr}_{S_m}\{\mathcal{C}_{\theta'} < \mathcal{B}_{\theta'}\} \cdot \mathbf{Pr}_{S_m}\{\theta_i = \theta'\}.$$

Since $\mathcal{B}_{\theta_i}$ is obtained using Lemma 4.1 (see Algorithm 1), and $\theta_i = \theta'$,

$$\mathbf{Pr}_{S_m}\{\mathcal{C}_{\theta_i} < \mathcal{B}_{\theta_i}\} = \mathbf{Pr}_{S_m}\{\mathcal{C}_{\theta'} < \mathcal{B}_{\theta'}\} < \frac{\delta}{k},$$

so we get,

$$\mathbf{Pr}_{S_m}\{\mathcal{C}_{\theta_i} < \mathcal{B}_{\theta_i}\}$$
$$= \int_0^1 d\theta' \mathbf{Pr}_{S_m}\{\mathcal{C}_{\theta'} < \mathcal{B}_{\theta'}\} \cdot \mathbf{Pr}_{S_m}\{\theta_i = \theta'\}$$
$$< \int_0^1 d\theta' \frac{\delta}{k} \cdot \mathbf{Pr}_{S_m}\{\theta_i = \theta'\}$$
$$= \frac{\delta}{k} \cdot \left( \int_0^1 d\theta' \mathbf{Pr}_{S_m}\{\theta_i = \theta'\} \right)$$
$$= \frac{\delta}{k}. \tag{4}$$

The following application of the union bound completes the proof,

$$\mathbf{Pr}_{S_m}\{\exists i : \mathcal{C}_{\theta_i} < \mathcal{B}_{\theta_i}\} \leq \sum_{i=1}^k \mathbf{Pr}_{S_m}\{\mathcal{C}_{\theta_i} < \mathcal{B}_{\theta_i}\} < \sum_{i=1}^k \frac{\delta}{k} = \delta.$$

$\square$

### 7.3 COMPLEXITY ANALYSIS

This section provides a brief complexity analysis of our method as well as the baselines (see Section 2). All baselines are lazy learners (analogous to nearest neighbors) in the sense that they require the entire source (detection-training) set for each detection decision they make. Using only a subset will result in sub-optimal performance. MMD is a permutation test (Gretton et al., 2012) that also employs a kernel. The complexity of kernel methods is dominated by the number of instances and, therefore, the time and space complexities of MMD are $O(d(m^2 + k^2 + mk))$ and $O(m^2 + k^2 + mk)$, respectively, where in the case of DNNs, $d$ is the dimension of the embedding or softmax layer used for computing the kernel. The KS test (Massey Jr, 1951) is a univariate test, which is applied on each dimension separately and then aggregates the results via a Bonferroni correction. Its time and space complexities are $O(d(m \log m + k \log k))$ and $O(d(m + k))$, respectively.

Our coverage-based detection algorithm is trained (only once) at $O(m \log m)$ time and $O(m)$ space complexities. Then, each detection round incurs $O(k+\log m)$ for both its time and space complexities. In practice, $dm^2 \gg dm \log m \gg \log m$, which makes our method significantly more efficient. For example, both baselines cannot process large "Google-scale" datasets, which our method can handle. A summary of these complexities appears in Table 1 (Section 4.2).

## 7.4 DETAILED EXPERIMENTS RESULTS

### 7.4.1 SYNTHETIC DATA

To gain some insight into the operation of our coverage-based detection algorithm we consider the following synthetic setting. A simple binary linear classifier was trained to discriminate between two 2D Gaussians, which are centered at $(-5, 0), (5, 0)$, respectively, whose covariance matrix is the identity. As the confidence-rate function (CF) we took the well-known softmax response (SR). For the shifted distribution we consider two cases where the modified distribution is defined by increasing or decreasing the distance between the Gaussians' centers. When the two Gaussians move closer, the coverage lower bound is expected to be violated. On the other hand, when they are further apart, the lower bound will not be violated. Thus, for this case, we introduce a symmetric coverage *upper* bound. This upper bound appears in Appendix 7.5. Hence, when the modified Gaussians are further apart, we expect that the upper bound will be violated. $G_0 \triangleq \{(-5, 0), (5, 0)\}$ are the centers of the Gaussians generating the detection-training data. $G_+ \triangleq \{(-6, 0), (6, 0)\}$ and $G_- \triangleq \{(-4, 0), (4, 0)\}$ represent two variations of $G_0$, where $G_+$ increases and $G_-$ decreases the distance between the centers.

We created two detection models by applying the training component of Algorithm 2 twice (to obtain the lower and upper bounds) with the following hyperparameters: $\delta = 0.001$ and detection-training set $S_m$ consisting of $m = 50,000$ samples generated from the Gaussians $G_0$ (25,000 from each). The set of desired coverages $\{c_i^*\}_{i=1}^{\lfloor \log_2 m \rfloor}$ is uniformly spread in the interval $[0, 1]$ (excluding its end points 0 and 1); thus, we have $\lfloor \log_2 50,000 \rfloor = 15$ desired coverages. To challenge our method in this setting, we generated samples from $G_+$, $G_-$ (expecting a violation of the upper and lower bounds, respectively), and also $G_0$ (as a sanity check that the bounds hold tightly).

We define the test windows $W_k^- \sim G_-, W_k^+ \sim G_+, W_k^0 \sim G_0$, each containing 25,000 samples from each of its two generating Gaussians (thus, $k = 50,000$). For example, in the case of $G_-$, we generate 25,000 samples from each Gaussian centered at $(-4, 0)$, and $(4, 0)$, respectively.

In Figure (4) we show the relationship between the empirical coverages $(\{\hat{c}_j^s(\theta_j, W_k^s)\}_{j=1}^{\lfloor \log_2 m \rfloor})$, the bounds $(\{b_j^*\}_{j=1}^{\lfloor \log_2 m \rfloor})$, and the desired coverages $(\{c_j^*\}_{j=1}^{\lfloor \log_2 m \rfloor})$, for each test window, $W_k^s, s \in \{-, +, 0\}$. Figures (4a) through (4d) show the results corresponding to the 'no-shift' and 'shift' cases, respectively.

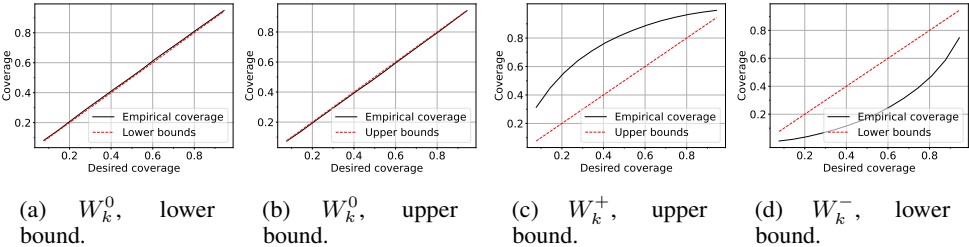

(a) $W_k^0$, lower bound.

(b) $W_k^0$, upper bound.

(c) $W_k^+$, upper bound.

(d) $W_k^-$, lower bound.

Figure 4: The empirical coverages and bounds as a function of the desired coverages, for each of the cases discussed in the text.

In particular, Figures (4c) and (4d) indicate that the windows $W_k^+$ and $W_k^-$ violate the upper and lower bounds, respectively. This behavior is expected, since the former $(W_k^+)$ should result in under-confident predictions, and the latter $(W_k^-)$, in over-confident predictions. The resulting p-values in both cases were nearly 0, which indicates a certain shift detection. The no-shift cases $(W_k^0)$ are shown in Figures (4a), (4b), from which it is evident that both bounds hold tightly (the p-value is nearly 1). Interestingly, when a shift occurs, see Figures 4c, 4d, the largest margin between the empirical coverage and the bound (in both the upper and lower cases) is obtained at the interior of the coverage range (e.g., around 0.7 coverage in the lower bound case, 4d). In other words, this synthetic experiment indicates that the detection effectiveness of our method is at its best in mid-range coverages. Specifically, Figure 5 shows the margin between the empirical coverages $(\{\hat{c}_j^s(\theta_j, W_k^s)\}_{j=1}^{\lfloor \log_2 m \rfloor})$, the bounds $(\{b_j^*\}_{j=1}^{\lfloor \log_2 m \rfloor})$ and the desired coverages $(\{c_j^*\}_{j=1}^{\lfloor \log_2 m \rfloor})$ for each

data window $W_k^s, s \in \{-, +, 0\}$. The no-shift case ($W_k^0$) results in a tight bound and, therefore, a very small margin (see Figures 5a and 5b). When shifts occur ($W_k^-, W_k^+$), the gap between the bounds and the empirical coverages is significantly larger and is not uniform (see Figures 5c and 5d).

We can thus infer that different data shifts would result in varying gap sizes, depending on the desired coverage being analyzed. In particular, the largest gap does not necessarily occur when $c^* = 1$, indicating that the detection power lies within lower coverages.

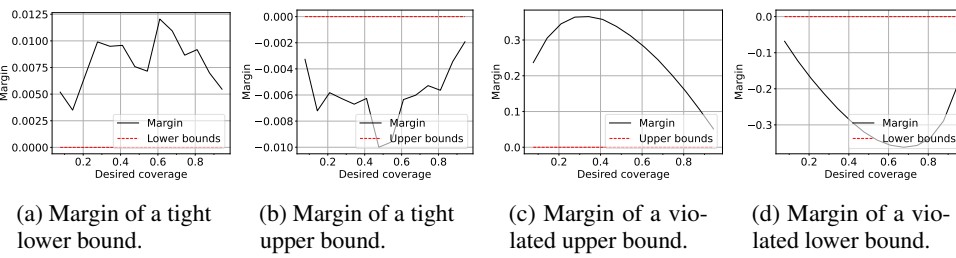

(a) Margin of a tight lower bound.

(b) Margin of a tight upper bound.

(c) Margin of a violated upper bound.

(d) Margin of a violated lower bound.

Figure 5: The margins between the bounds and the actual coverage.

### 7.4.2 CIFAR-10 Experiments

The score for each window $W_k$, of size $k$, as well as the margin between our best method (Ours-Ent) and the best baseline (KS-BBSD-S) considering the ResNet-18 and ResNet-50 architectures, can be found in Figures 6, 7, respectively.

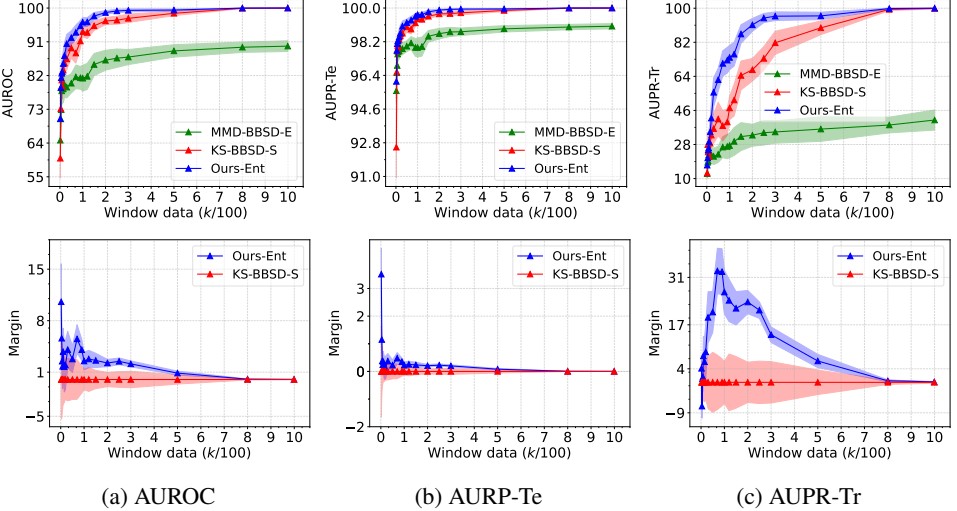

(a) AUROC        (b) AURP-Te        (c) AUPR-Tr

Figure 6: CIFAR-10 Detection results using ResNet-18. The metric scores as a function of window size (upper), and the margin between our best method and the best baseline, (lower). The margin is the difference between the metrics (AUROC, AUPR-Te, AUPR-Tr) scores of Ours-Ent and KS-BBSD-S. One-$\sigma$ error-bars are shadowed.

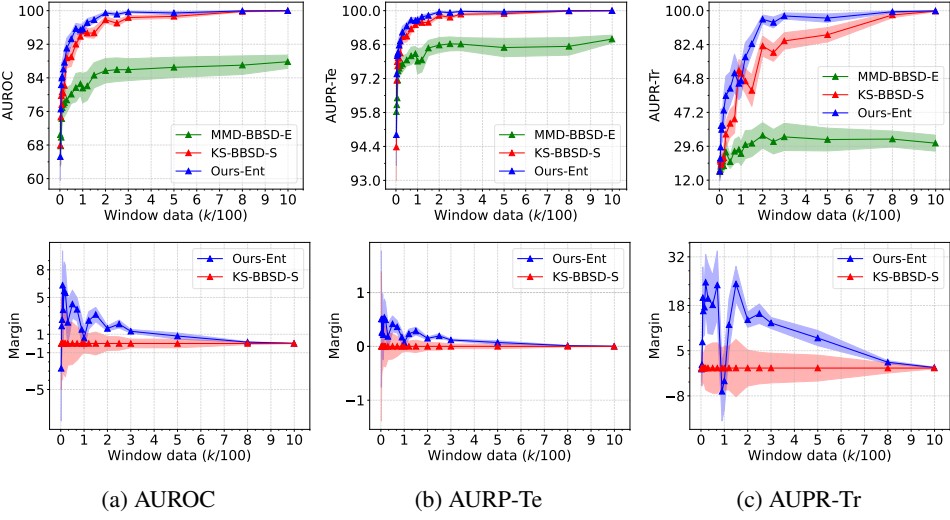

(a) AUROC        (b) AURP-Te        (c) AUPR-Tr

Figure 7: CIFAR-10 Detection results using ResNet-50. The metric scores as a function of window size (upper), and the margin between our best and the best baseline, (lower). The margin is the difference between the metrics (AUROC, AUPR-Te, AUPR-Tr) scores of Ours-Ent and KS-BBSD-S. One-$\sigma$ error-bars are shadowed.

### 7.4.3 IMAGENET EXPERIMENTS

Regarding the ImageNet experiment, the score for each window $W_k$, of size $k$, as well as the margin between our best method (Ours-Ent) and the best baseline (KS-BBSD-S), considering the EfficientNet-B0 architecture, is given in Figures 8, 9, 10, for $p = 1$, $p = 2/3$, and $p = 1/3$, respectively.

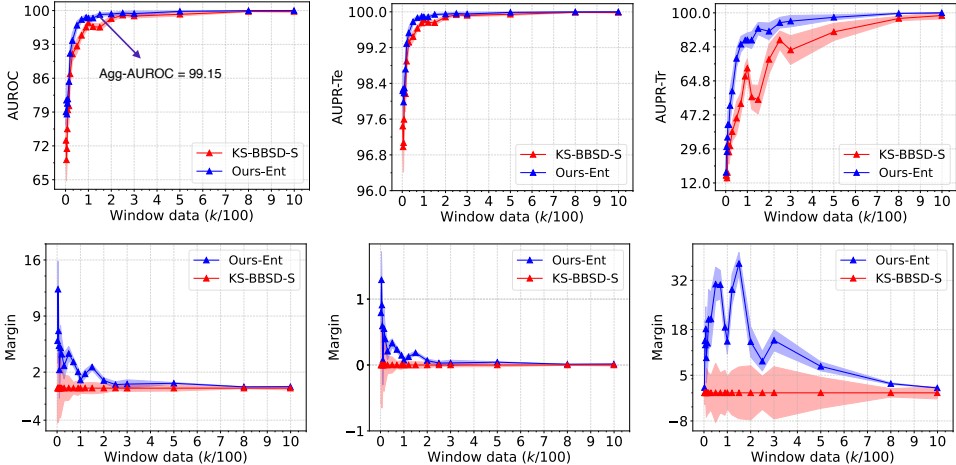

Figure 8: ImageNet Detection results using EfficientNet-B0 with $p = 1$. The metric scores as a function of window size (upper), and the margin between our best and the best baseline, (lower). The margin is the difference between the metrics (AUROC, AUPR-Te, AUPR-Tr) scores of Ours-Ent and KS-BBSD-S. One-$\sigma$ error-bars are shadowed.

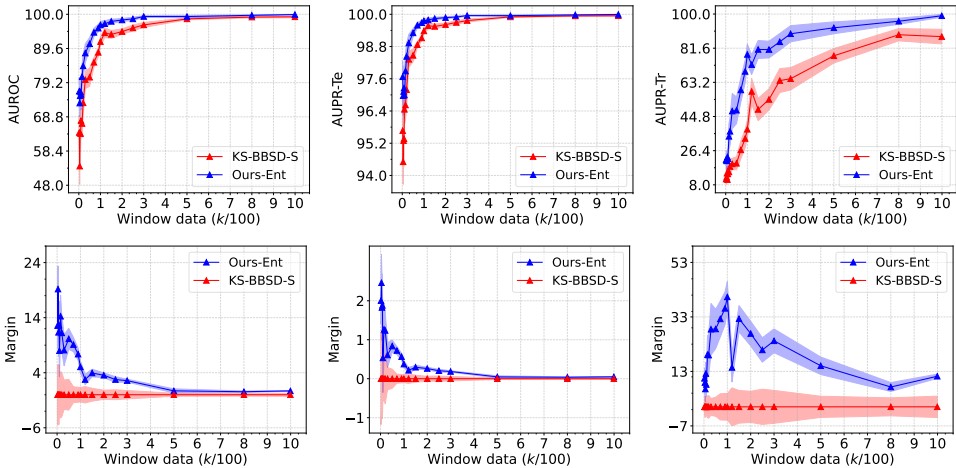

Figure 9: ImageNet Detection results using EfficientNet-B0 with $p = 2/3$. The metric scores as a function of window size (upper), and the margin between our best and the best baseline, (lower).The margin is the difference between the metrics (AUROC, AUPR-Te, AUPR-Tr) scores of Ours-Ent and KS-BBSD-S. One-$\sigma$ error-bars are shadowed.

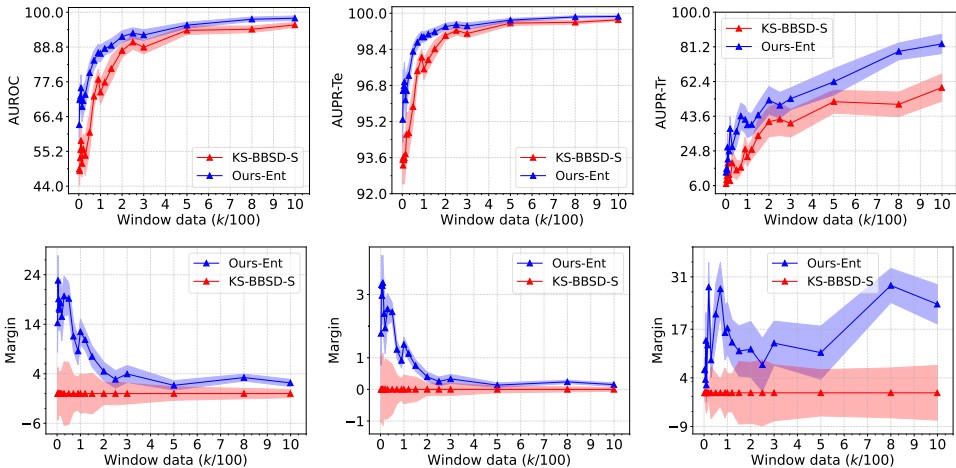

Figure 10: ImageNet Detection results using EfficientNet-B0 with $p = 1/3$. The metric scores as a function of window size (upper), and the margin between our best method and the best baseline, (lower). The margin is the difference between the metrics (AUROC, AUPR-Te, AUPR-Tr) scores of Ours-Ent and KS-BBSD-S. One-$\sigma$ error-bars are shadowed.

### 7.5 SYMMETRIC UPPER BOUND

Here we develop an upper coverage bound.

For a classifier $f$, a detection-training sample $S_m \sim P^m$, a confidence parameter $\delta > 0$, and a desired coverage $c^* > 0$, our goal is to use $S_m$ to find a $\theta$ value (which implies a selection function $g_\theta$), such that the coverage satisfies,

$$\mathbf{Pr}_{S_m}\{c(\theta, P) > c^*\} < \delta. \tag{5}$$

This means that *over coverage* should occur with probability of at most $\delta$.

The pseudo code of the algorithm that finds the optimal coverage upper bound (with confidence $\delta$), appears in Algorithm 3.

---

**Algorithm 3:** *Selection with guaranteed coverage - Upper bound*

---

1 **Input:** train set: $S_m$, confidence-rate function: $\kappa_f$, confidence parameter $\delta$, target coverage: $c^*$.
2 Sort $S_m$ according to $\kappa_f(x_i)$, $x_i \in S_m$ (and now assume w.l.o.g. that indices reflect this ordering).
3 $z_{\min} = 1$, $z_{\max} = m$
4 **for** $i = 1$ **to** $k = \lceil \log_2 m \rceil$ **do**
5 $\quad z = \lceil (z_{\min} + z_{\max})/2 \rceil$
6 $\quad \theta_i = \kappa_f(x_z)$
7 $\quad$ Calculate $\hat{c}_i(\theta_i, S_m)$
8 $\quad$ Solve for $b_i^*(m, m \cdot \hat{c}_i(\theta_i, S_m), \frac{\delta}{k})$ {see Lemma 4.1}
9 $\quad$ **if** $b_i^*(m, m \cdot \hat{c}_i(\theta_i, S_m), \frac{\delta}{k}) \geq c^*$ **then**
10 $\quad\quad z_{\min} = z$
11 $\quad$ **else**
12 $\quad\quad z_{\max} = z$
13 $\quad$ **end if**
14 **end for**
15 **Output:** bound: $b_k^*(m, m \cdot \hat{c}_k(\theta_k, S_m), \frac{\delta}{k})$, threshold: $\theta_k$.

---

Similarly to SGC (Algorithm 1), Algorithm 3 uses Lemma 7.1. The proof of Lemma 7.1 can be easily deduced from the proof of Lemma 4.1.

**Lemma 7.1.** *Let P be any distribution and consider a CF threshold $\theta$ with a coverage of $c(\theta, P)$. Let $0 < \delta < 1$ be given and let $\hat{c}(\theta, S_m)$ be the empirical coverage w.r.t. the set $S_m$, sampled i.i.d. from P. Let $b^*(m, m \cdot \hat{c}(\theta, S_m), \delta)$ be the solution of the following equation:*

$$\arg\min_b \left( \sum_{j=0}^{m \cdot \hat{c}(\theta, S_m)} \binom{m}{j} b^j (1-b)^{m-j} \leq \delta \right). \tag{6}$$

*Then,*

$$\mathbf{Pr}_{S_m}\{c(\theta, P) > b^*(m, \hat{c}(\theta, S_m), \delta)\} < \delta. \tag{7}$$

### 7.6 COMPARISON BETWEEN OUR TWO PROPOSED METHODS

The following Figure (11) compares between our two proposed methods, Ours-Ent, Ours-SR, for a contamination value of $p = 2/3$. Figure 11(left) - shows the AUROC metric as a function of window size, Figure 11(right) - shows the margin between the two methods.

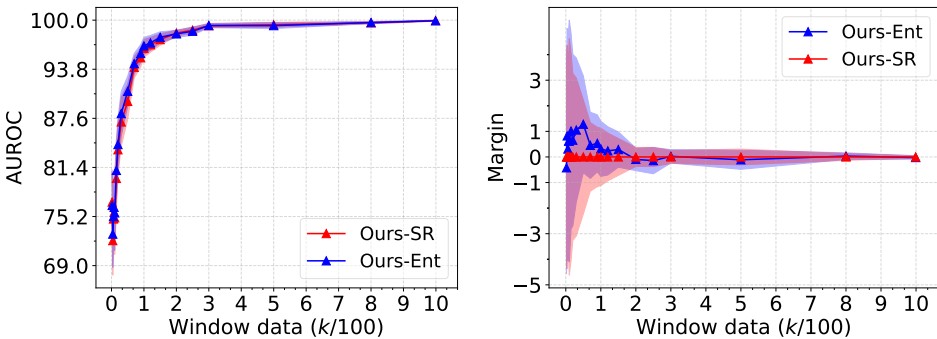

Figure 11: Comparison between Ours-Ent, and Ours-SR. One-$\sigma$ error-bars are shadowed.

## 7.7 PERFORMANCE PER DISTRIBUTION SHIFT - IMAGENET

The following graphs demonstrate representative results regarding the detection performance per distribution shift. Specifically, the first row of Figure 12 represents the AUROC metric where the contamination value is two thirds ($p = 2/3$), for the following diverse shifts separately: ImageNet C-severity 1, Imagenet A, rotation of 40 degrees, rotation of 180 degrees and the best considered adversarial attack, CW. The second row of Figure 12 represents the margin between our best method (Our-Ent) and the best baseline (KS-BBSD-S).

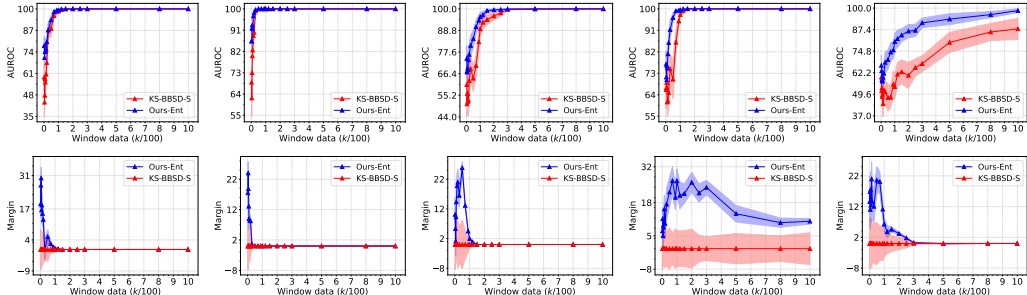

Figure 12: ImageNet Partial detection results, for individual distribution shifts. One-$\sigma$ error-bars are shadowed.

## 7.8 BROADER EXPLANATION OF LEMMA 4.1

Lemma 4.1 gives (the tightest possible) generalization numerical bound of a coverage, given a test sample, $S_m$, and a confidence parameter $\delta$. Equation 2 solves for the inverse binomial, which stands for the probability, $b$, such that an event with at most $m \cdot \hat{c}(\theta, S_m)$ "successes" will equal $1 - \delta$. Since the inverse binomial is a monotonic decreasing function of $b$, we look for the minimum probability. Lemma 4.1 states, that the solution of Equation 2, $b^*$, is a lower bound, which holds with probability $1 - \delta$, and therefore, the probability of violating this bound is $\delta$. This is stated in Equation 3.

