# OpenReview forum: "Distribution Shift Detection for Deep Neural Networks"
_ICLR.cc/2023/Conference — Submitted to ICLR 2023_

### Official Review · Reviewer_TC5F · 2022-10-20

**Confidence:** 3
**Correctness:** 4
**Technical Novelty And Significance:** 4
**Empirical Novelty And Significance:** 3
**Recommendation:** 6

**Clarity, Quality, Novelty And Reproducibility:**

## Clarity

Other than some minor issues above, everything is clear. I would specifically like to see an intuitive explanation of equations 2-3 added for clarity.

## Quality

The quality is high, in my opinion

## Novelty

To my knowledge, the paper is sufficiently novel

**Strength And Weaknesses:**

## Strengths

- The method is well motivated and sound.
- The method outperforms previous methods with impressive gains in computational complexity.

## Weaknesses

- Do the metrics outlined in section 2 such as MMD and KS-BBSD have the implcit assumption that the data is coming in from an unbiased sample of the Q distribution? For instance, would everything break down if the test time environment made predictions by streaming many instances of a single class at once?

- I ask the question above, because the testing data being an unbiased sample from the data seems to be an implicit assumption of the method, and one which might not be realisitic for an entire range of models in practice, which may hinder its usefulness. I realize this is a consequence of the chosen problem setting, but I think at the very least, this should be stated as a limitation which (I think) applies to the selected baselines as well.

- Page 4: "more precisely one minus the entropy..." Entropy of a softmax $h$ is in the range $[0, \infty]$. So why would one want to look at $1 - h$? Does this mean to only look at the class prediction index as a single Bernoulli variable?

- The ImageNet experiments in Table 3 aggregate over all corruptions of the different imagenet datasets. It would be nice to see breakdowns of the performances on individual datasets as well (maybe in the appendix) to see if any patterns emerge between the different types of datasets.

- The intuition around equation 2 and 3 could be expanded, maybe in the appendix. The understanding of the whole paper rests on these two equations, and they were both hard to digest at first without great effort for me. I think it would be easier for most readers to grasp if equations 2-3 could be justified with a non-rigorous, but intuitive explanation.

## Minor

- Page 2: "As shown here, the KS-BBSD..." as shown where? This sentence is unclear.
- is the sorting in algorithm 1 ascending or descending? Or does it not matter?
- Section 4.2: $(b^*_j, \theta_j)$ is said to be a threshold and a bound, respectively, but the algorithm has those terms reversed. Shouldn't it say "bound and a threshold, resepctively?"

**Summary Of The Paper:**

The authors propose a method for identifying when a distirbutional shift from the training data has been received in a DNN. The method considers a window of recent instances as opposed to identifying single OOD instances.

**Summary Of The Review:**

Overall, I think this is a solid work, which gives an adequate contribution. I would still like to see the concerns raised above be addressed.

---

> ### Author Response · Authors · 2022-11-11
> **General Response**
>
> Thanks for the positive feedback. We are committed to addressing all your questions and concerns.
>
> *Do the metrics outlined in section 2 such as MMD and KS-BBSD have the implicit assumption that the data is coming in from an unbiased sample of the Q distribution? For instance, would everything break down if the test time environment made predictions by streaming many instances of a single class at once?*
>
> We are not sure we understand your question correctly.
> What we do understand (and this is also related to your next comment) is that you are concerned about bad distributions that will fail our method. In regards to our method, it is theoretically possible to craft a bad attacking distribution Q that will exhibits exactly the same risk-coverage (RC) curve (see definitions in [1], Section 2) as the one P has (we believe this is the only way to systematically fail our method). However, it is highly unlikely to encounter such distributions naturally. We believe that also the baselines can be failed by a malicious construction of some sort.
> Having said that, we will be glad if you clarify the question in case we have missed the point.
>
> *I ask the question above, because the testing data being an unbiased sample from the data seems to be an implicit assumption of the method, and one which might not be realisitic for an entire range of models in practice, which may hinder its usefulness. I realize this is a consequence of the chosen problem setting, but I think at the very least, this should be stated as a limitation which (I think) applies to the selected baselines as well.*
>
> See our response to the previous question.
>
> *Page 4: "more precisely one minus the entropy..." Entropy of a softmax $h$ is in the range $[0,\infty]$, so why would one want to look at 1 - h? Does this mean to only look at the class prediction index as a single Bernoulli variable?*
>
> Indeed, we did not specify how we calculate the entropy of the probability vector. We used base $C$ for the logarithm. That is, $H(p) = -\sum_{i=1}^C p(i)\log_C p(i)$.
> This ensures that $0 \leq H(p) \leq 1$.
>
> *The ImageNet experiments in Table 3 aggregate over all corruptions of the different imagenet datasets. It would be nice to see breakdowns of the performances on individual datasets as well (maybe in the appendix) to see if any patterns emerge between the different types of datasets.*
>
> Full analysis containing performance per distribution shift class will be added to the revised version. As a step toward including all these detailed results we already added Appendix 7.7 (highlighted in blue), containing Figure 12, which will be included in the revised version. Figure 12 presents the AUROC metric where $p=2/3$, for each of the following shifts separately: ImageNet C-severity 1, Imagenet A, rotation of 40 degrees, rotation of 180 degrees and the best considered adversarial attack, CW.
> Observe that for all window sizes, the blue line dominates the red line, namely, our method outperforms the best baseline.
> The same is true for the vast majority of other shifts as well as for the two other contamination values, $p=1, 1/3$.
>
> *The intuition around equation 2 and 3 could be expanded, maybe in the appendix. The understanding of the whole paper rests on these two equations, and they were both hard to digest at first without great effort for me. I think it would be easier for most readers to grasp if equations 2-3 could be justified with a non-rigorous, but intuitive explanation.*
>
> Thanks for pointing this out. We can clarify those two equations as follows, firstly, we add (in the revised version) an initial explanation regarding our method at the beginning of Section 4 (changes are highlighted in blue). Secondly, we add Appendix 7.8, which provides an intuitive explanation of the two equations (also highlighted in blue).
>
> *Page 2: "As shown here, the KS-BBSD..." as shown where? This sentence is unclear.*
>
> Fixed - See changes in the revised version highlighted in blue.
>
> *is the sorting in algorithm 1 ascending or descending? Or does it not matter?*
>
> We sort the values in ascending order.
>
> *Section 4.2: $b^\ast, \theta_j$, is said to be a threshold and a bound, respectively, but the algorithm has those terms reversed. Shouldn't it say "bound and a threshold, respectively?"*
>
> Thanks for catching this mistake. We fixed this issue (highlighted in blue).
>
>
> [1] - On the Foundations of Noise-free Selective Classification Ran El-Yaniv and Yair Wiener 2010

---

> > ### Comment · Reviewer_TC5F · 2022-11-25
> > **Clarification**
> >
> > Thank you for answering my questions, I will clarify the point below.
> >
> > > We are not sure we understand your question correctly. What we do understand (and this is also related to your next comment) is that you are concerned about bad distributions that will fail our method. In regards to our method, it is theoretically possible to craft a bad attacking distribution Q that will exhibits exactly the same risk-coverage (RC) curve (see definitions in [1], Section 2) as the one P has (we believe this is the only way to systematically fail our method). However, it is highly unlikely to encounter such distributions naturally. We believe that also the baselines can be failed by a malicious construction of some sort. Having said that, we will be glad if you clarify the question in case we have missed the point.
> >
> > For example, consider the normal training process with your method, now consider that at test time, for some reason or another, the data arrives in a sorted order so that it is sorted by class. In this case, we can assume that the window $W_k$ will be completely filled with one class which likely invalidates the method, right?
> >
> > So if we consider the coverage was tuned with an unbiased sample from training data $P$, then the model may still fail even if $P=Q$, but the samples from $Q$ are biased. I don't mean to imply that your method should protect against this, or that the baselines don't also suffer from this issue, I only bring this up because I think this should be listed as a limitation due to the base assumption of the model.

---

### Official Review · Reviewer_Uqk1 · 2022-10-25

**Confidence:** 2
**Correctness:** 4
**Technical Novelty And Significance:** 2
**Empirical Novelty And Significance:** 3
**Recommendation:** 5

**Clarity, Quality, Novelty And Reproducibility:**

The paper describes the considered problem and clarifies its scope very well. The experiments are of high quality. But the novelty and clarity of the methods still need to be addressed.

I also have the following questions while reading the paper:
- Why do you choose $log_2 m$ as the number of target coverages?
- How do you solve the *equation* in Lemma 4.1 and what is the complexity? What do you mean by *equation*? Curiously, what is an argmin of an inequality?
- When the majority of the window contains normal data, how do you determine the window as a distribution shift or normal samples?
- Because some results in the tables are quite close, it would be good to report the error bar as well to explain the significance.
- When I was reading the paper, I didn’t feel the *selective classifier* fit into the paper. I could still get a rough picture without it. Or if I only knew the confidence function, I could still get the idea.


**Strength And Weaknesses:**

Strengths:
- The paper does a good work of explaining its scope.
- The experiments are of high quality. Experiments are comprehensive and evaluate a lot of datasets.

Weaknesses:
- My main concern is the novelty of the paper. The paper proposes a new method for distribution shift detection. However, based on the description of the authors, the proposed method seems like a quick combination of two works (Langford & Schapire, 2005 and Geifman & El-Yaniv, 2017) without much modification. Maybe state clearly which efforts you made to adapt the two referenced works to the current task.
- I feel the main idea and the motivation of the proposed method can be better explained intuitively with plain language. So the method can get better clarity. For example, “... find better summary statistics of the ID data inspired by generalization coverage bound…” (just an example, please ignore my terminology if I’m wrong).


**Summary Of The Paper:**

The paper focuses on improving distribution shift detection given a pre-trained deep model. It proposes a coverage-based statistic and uses the statistic to detect distribution shifts. The statistics are proven to have a high-probability coverage of the in-distribution data. The detection step is through a two-sample t-test. Experiments on CIFAR and ImageNet show the proposed method outperforms existing baselines.

**Summary Of The Review:**

The paper proposes a novel distribution shift detection method for deep models. The authors seem to quickly combine two works (Langford & Schapire, 2005 and Geifman & El-Yaniv, 2017) for their proposed solution. The novelty of the work is not enough without further addressing this.

---

> ### Author Response · Authors · 2022-11-11
> **General Response**
>
> Thank you for the feedback and suggestions. We are committed to address all your questions or concerns.
>
> *My main concern is the novelty of the paper. The paper proposes a new method for distribution shift detection. However, based on the description of the authors, the proposed method seems like a quick combination of two works (Langford and Schapire, 2005 and Geifman and El-Yaniv, 2017) without much modification. Maybe state clearly which efforts you made to adapt the two referenced works to the current task.*
>
> The technical underpinnings of our paper are not the main novelty. The main innovation in this paper is the use of Lemma 4.1 to detect distributional shifts from unlabeled data. Further, we demonstrate by extensive experimentation that this theoretically motivated idea leads to state-of-the-art performance in detecting distribution shifts.
>
>
> *I feel the main idea and the motivation of the proposed method can be better explained intuitively with plain language. So the method can get better clarity. For example, “... find better summary statistics of the ID data inspired by generalization coverage bound…” (just an example, please ignore my terminology if I’m wrong).*
>
> Thanks for pointing this out. We revised the intuition provided in Section 4. Changes are highlighted in blue.
>
> *Why do you choose $log_2 m$ as the number of target coverages?*
>
> $log_2 m$ was a natural choice for a monotone, yet sub-linear number of target coverages. This number was not optimized, and is fixed for all experiments.
>
> *How do you solve the equation in Lemma 4.1 and what is the complexity? What do you mean by equation? Curiously, what is an argmin of an inequality?*
>
> The inverse binomial is a monotonic decreasing function. Therefore, the solution is given by requiring an equality.
> We solve the equation by performing a binary search over the value of the binomial, and equating it to $1-\delta$.
> The complexity is ceil($\log_2 \frac{1}{\epsilon}$), where $\epsilon$ is the tolerance (the difference between the value of the binomial and the desired value). We used $\epsilon = e^{-7}$, so 11 iterations are required.
>
>
> *When the majority of the window contains normal data, how do you determine the window as a distribution shift or normal samples?*
>
> We compare (by conducting a t-test) the empirical coverage of the (test) window, to the coverage bound calculated based on in-distribution data. We determine if a distribution shift occurs via a p-value, and a user defined significant level $\alpha$. In our experiments, we use $1- p\_{value}$ as a score for a (test) window being considered in-distribution, and our evaluation metrics are via this score function.
>
> *Because some results in the tables are quite close, it would be good to report the error bar as well to explain the significance.*
>
> We have error-bars reported for **each** window size we experimented on. We refer you to Appendix 7.4.2, Figures 6,7 for the CIFAR10 experiments, and to Appendix 7.4.3, Figures 8,9,10 for the ImageNet experiments. Our methods outperform the baselines with statistical significance in the vast majority of window sizes.
>
> *When I was reading the paper, I didn’t feel the selective classifier fit into the paper. I could still get a rough picture without it. Or if I only knew the confidence function, I could still get the idea.*
>
> Thanks for pointing this out. We will rethink mentioning the selective classifier in the final version of the paper.

---

> > ### Author Response · Authors · 2022-11-26
> > **Relating to our response**
> >
> > Dear reviewer Uqk1,
> > Hopefully, we have addressed your concerns properly. We would greatly appreciate it if you could confirm that or raise any unresolved issues.

---

### Official Review · Reviewer_Mwu4 · 2022-10-25

**Confidence:** 3
**Clarity, Quality, Novelty And Reproducibility:** The paper is mostly clear and well-or…
**Correctness:** 3
**Technical Novelty And Significance:** 3
**Empirical Novelty And Significance:** 2
**Recommendation:** 6

**Strength And Weaknesses:**


Strengths
- The method considers the window-based settings, which is relatively less explored in the literature.
- The method is theoretically motivated and performs quite well in practice.

Weaknesses
- Lack empirical comparison with single-instance OOD detection methods. The paper mentions that "Single-instance methods are trivially applicable to a window". However, no methods are compared. What are the practical implications of "not considering population statistics over the window"? I wonder if authors provide empirical comparisons with representative methods in the literature, such as MaxLogit score [1] derived from the logit space and KNN score [2] derived from the feature embeddings.
- Unlike the majority of instance-based methods in the literature, the proposed method is limited to window-based inputs. e.g., it might be better to change the title to indicate the specific set up.


[1] Hendrycks et al., Scaling Out-of-Distribution Detection for Real-World Settings, ICML 2022
[2] Sun et al., Out-of-Distribution Detection with Deep Nearest Neighbors, ICML 2022

**Summary Of The Paper:**

The paper proposes a new out-of-distribution detection method when sampled are provided within a given window.  The core algorithm aims to find optimal coverage lower bound and the detection threshold given confidence parameter, and target coverage. The authors provide theoretical analysis and empirical studies on small-scale and large-scale datasets.

**Summary Of The Review:**

The paper is theoretically motivated, sample-efficient compared to the baselines, and empirically performs well. However, the empirical evaluation can be further expanded.

---

> ### Author Response · Authors · 2022-11-11
> **General Response**
>
> Thank you for the positive feedback and suggestions. We are committed to address all your questions or concerns.
>
> *Lack empirical comparison with single-instance OOD detection methods. The paper mentions that "Single-instance methods are trivially applicable to a window". However, no methods are compared. What are the practical implications of "not considering population statistics over the window"? I wonder if authors provide empirical comparisons with representative methods in the literature, such as MaxLogit score [1] derived from the logit space and KNN score [2] derived from the feature embeddings.*
>
> The window-based population technique we present is able to achieve 100 percent AUROC score across all distribution shifts considered (rotations, Gaussian noise, adversarial attacks, etc...) when using a sufficiently large window size (e.g., using ImageNet as the source distribution). Moreover, its detection score is monotonically increasing as a function of the window size. This is depicted in Figure 2 in our paper, which shows the detection score for each window size, with one-sigma error-bar shadowed. Single-instance detection methods rarely achieve such a high scores. This argues for the benefit of using population based detection techniques.
> More specifically, Table 1 in [1], and Table 4 in [2] (which you mentioned), report the AUROC score for OOD detection for ImageNet as the in-distribution data set. MaxLogits ([1]) achieves an average AUROC score of 87.2, and the average KNN score ([2]) over several datasets is 90.91.
> Having said that, and as we mention in the concluding remarks section, a very intriguing question is whether one can
> benefit from using both single instance techniques (such as MaxLogits) combined with population-based methods to improve detection performance on smaller windows.
>
> *Unlike the majority of instance-based methods in the literature, the proposed method is limited to window-based inputs. e.g., it might be better to change the title to indicate the specific set up.*
>
> We agree with the reviewer's comment. We changed the title to 'Window-based distribution shift detection for deep neural networks' in the revised version. Changes are highlighted in blue.

---

> > ### Author Response · Authors · 2022-11-26
> > **Relating to our response**
> >
> > Dear reviewer Mwu4,
> > Hopefully, we have addressed your concerns properly. We would greatly appreciate it if you could confirm that or raise any unresolved issues.

---

### Official Review · Reviewer_RPBQ · 2022-11-11

**Confidence:** 4
**Correctness:** 3
**Technical Novelty And Significance:** 3
**Empirical Novelty And Significance:** 3
**Recommendation:** 6

**Clarity, Quality, Novelty And Reproducibility:**

This paper is mostly well-organized and written. The paper is somehow novel, which investigates the distribution shift detection from a multi-instance angle, which also gives a nice discussion of previous work.

**Strength And Weaknesses:**

Strength:

- Important problem
- Group-based distribution shift detection is less investigated, the proposed techniques are Sound and feasible solution
- Some formal guarantees of the bound are analyzed
- Extensive evaluation to demonstrate the potential usefulness
- Promising results

Weakness:

- Evaluation is only performed on image data for classification tasks.
- The scenarios of the distribution shifts are mostly simulated, not real scenarios from the real world.
- Unclear about the insights of “k” selection and its relation to the population size of training data or the original data distribution P.
- Unclear how to extend to streaming data cases.

**Summary Of The Paper:**

This paper proposes a distribution shift detection method, with the information of k-window data instances. The paper proposes a coverage-based method with a guaranteed bound, based on this, a further detection algorithm is proposed for efficient processing with the advantages of algorithmic complexity in terms of both space and time. The evaluation results also demonstrate the potential of the proposed technique.

**Summary Of The Review:**

Overall, I enjoyed reading this paper, the proposed techniques should be sound and feasible. The evaluation is relatively comprehensive on the case of CIFAR and ImageNet. In addition, the authors also provide a formal analysis of the bound analysis,  as well as the detection complexity analysis of the proposed algorithms.

Even though, the paper still posts a few concerns that the authors could consider for further enhancement.

- The proposed methods rely on quite a few hyper-parameters, I would recommend add more discussion on their insights, the heuristic to choose them, as well as evaluating the impact of such selection.

- During the evaluation, the authors mostly compared with KS and MMD methods, and simply believed that the single-instance-based methods e.g., for OoD detection would not work, which is not fully convincing to me. I would highly recommend adding comparisons with some SOTA instance-based OoD/Uncertainty detection liked methods, under the window k, for comparative analysis. This could be critical from my perspective, especially if the instance-based-OoD detection already works quite well, which is also efficient in some cases, e.g., under what conditions.

- Regarding the evaluation scenario, I understand authors might try hard to try simple scenarios to simulate distribution shift, but the evaluated scenarios are quite artificial, which might be quite far away from real-world cases. I would recommend authors (1) considering some real-world cases, (2) as well as evaluating on more diverse tasks.

For example, the following paper could be a good reference.

**WILDS: A Benchmark of in-the-Wild Distribution Shifts**

***Pang Wei Koh, Shiori Sagawa, Henrik Marklund, Sang Michael Xie, Marvin Zhang, Akshay Balsubramani, Weihua Hu, Michihiro Yasunaga, Richard Lanas Phillips, Irena Gao, Tony Lee, Etienne David, Ian Stavness, Wei Guo, Berton Earnshaw, Imran Haque, Sara M Beery, Jure Leskovec, Anshul Kundaje, Emma Pierson, Sergey Levine, Chelsea Finn, Percy Liang***
 *Proceedings of the 38th International Conference on Machine Learning*
, PMLR 139:5637-5664, 2021.

---

> ### Author Response · Authors · 2022-11-13
> **General Response**
>
> Thank you for the positive feedback and suggestions. We are committed to address all your questions or concerns.
>
> *The scenarios of the distribution shifts are mostly simulated, not real scenarios from the real world.*; *Regarding the evaluation scenario, I understand authors might try hard to try simple scenarios to simulate distribution shift, but the evaluated scenarios are quite artificial, which might be quite far away from real-world cases. I would recommend authors (1) considering some real-world cases, (2) as well as evaluating on more diverse tasks.*
>
> While we agree that it is hard to simulate ``real world'' scenarios (and, moreover, such scenarios are task dependent), in our ImageNet experiments, we used ImageNet-O and ImageNet-A (and many more datasets, see Section 5.1.2) as out-of-distribution datasets.
> According to the authors of the paper introducing those sets [2], these represent "real-world, unmodified, and naturally occurring examples that cause machine learning model performance to significantly degrade" (Abstract).
> Finally, thanks for pointing out the paper "Wilds: A benchmark of in-the-wild distribution shifts". We will make sure to cite it and mention that future work should consider it. See changes in Section 6, highlighted in blue.
>
> *Unclear about the insights of “k” selection and its relation to the population size of training data or the original data distribution P.*
>
> To span $0 \leq k \leq 1000$ we used the following values, $k=\{ 2,4,6,8,10,20 ,30,40,50,60,80,100,120, 150, 200, 250, 300, 500, 800, 1000\}$.
> We sampled lower values of $k$ more densely because large $k$ gave rise to nearly perfect performance.
>
> *Unclear how to extend to streaming data cases.*
>
> We are not sure we understand your question.
> Are you asking how quickly our detector will detect a shift within a window (i.e., as the contamination percentage grows)?
> If this is the question, we have the following answer.
> Please observe Figures~8,9,10 in Appendix 7.4.3. These figures demonstrate all evaluation metrics, as a function of the window size, for three contamination percentages, $1, 2/3, 1/3$. As might be anticipated, it is harder to detect at lower contamination ratios.
> That being said, we would appreciate a clarification if we missed the point.
>
> *The proposed methods rely on quite a few hyper-parameters, I would recommend add more discussion on their insights, the heuristic to choose them, as well as evaluating the impact of such selection.*
>
> Our algorithm requires three hyper-parameters: $\delta, \kappa$,  $\log_2m$ target coverages.
>
> Their role is as follows:
>
> $\delta$: this parameter holds for the probability that equation~(3) doesn't hold. Ideally we want a small value and we fixed it to $\delta=0.001$ through all experiments without any optimization.
>
> $\kappa$: we experimented with both softmax-response (SR) and Entropy (Ent) as $\kappa$. We provide a short analysis/comparison of those two confidence functions in Appendix 7.6. Ent seems to perform slightly better.
>
>  $\log_2m$ target coverages: we chose $\log_2 m$ target coverage uniformly spread in the interval $[0,1]$. The choice of $\log_2 m$ was a natural choice for a monotone, yet sub-linear number of target coverages. This number was not optimized, and is fixed for all experiments.
> In general, there is no danger in taking a larger number of target coverages and performance will tend to improve with more values.
>
> *During the evaluation, the authors mostly compared with KS and MMD methods, and simply believed that the single-instance-based methods e.g., for OoD detection would not work, which is not fully convincing to me. I would highly recommend adding comparisons with some SOTA instance-based OoD/Uncertainty detection liked methods, under the window k, for comparative analysis. This could be critical from my perspective, especially if the instance-based-OoD detection already works quite well, which is also efficient in some cases, e.g., under what conditions.*
>
> The window-based population technique we present is able to achieve 100 percent AUROC score across all distribution shifts considered (rotations, Gaussian noise, adversarial attacks, etc...) when using a sufficiently large window size (e.g., using ImageNet as the source distribution). Moreover, its detection score is monotonically increasing as a function of the window size. This is depicted in Figure 2 in our paper, which shows the detection score for each window size, with one-sigma error-bar shadowed. Single-instance detection methods rarely achieve such high scores.
> Having said that, and as we mention in the concluding remarks section, an instance-based-OoD is valuable for small windows, and a very intriguing question is whether one can benefit using both single instance techniques combined with population-based methods to improve detection performance on smaller windows.
>
> [2] Natural Adversarial Examples Dan Hendrycks, Kevin Zhao, Steven Basart, Jacob Steinhardt, Dawn Song, CVPR 2021

---

> > ### Author Response · Authors · 2022-11-26
> > **Relating to our response**
> >
> > Dear reviewer RPBQ,
> > Hopefully, we have addressed your concerns properly. We would greatly appreciate it if you could confirm that or raise any unresolved issues.

---

### Decision · Program_Chairs · 2023-01-20

**Decision:**

Reject

**Justification For Why Not Higher Score:**

see above

**Justification For Why Not Lower Score:**

N/A

**Metareview: Summary, Strengths And Weaknesses:**

The paper proposes a distribution shift detection method, considering a window-based approach (i.e. multiple instances are considered simultaneously) compared to a single instance setting. The reviewers agree that the problem itself is interesting and has been less studied than single instance detection (as such, there is some novelty). However, the applicability of such principle in practice has also been raised.

The main limitation of the work is its missing comparison to single instance based methods. The problem has been raised by the reviewers but has not been appropriately addressed by the authors in their feedback (i.e. adding corresponding experiments). Given this, the consensus was a "borderline; slightly below acceptance".

**Summary Of Ac-Reviewer Meeting:**

There was a live meeting between AC and the reviewers. As mentioned in the summary above, the main limitation of the work is its missing comparison to single instance methods. Since the paper has been evaluated as borderline overall, this point has led to the decision of "borderline; slightly below acceptance". However, the score can also be "bumped up" if there is space in the program.